# Genomic Biomarkers of Survival in Patients with Adenocarcinoma of the Uterine Cervix Receiving Chemoradiotherapy

**DOI:** 10.3390/ijms21114117

**Published:** 2020-06-09

**Authors:** Ying-Chun Lin, Yu-Chia Chen, Rui-Yun Chen, Yi-Xuan Huang, Siang-Jyun Tu, Ji-An Liang, Yao-Ching Hung, Lian-Shung Yeh, Wei-Chun Chang, Wu-Chou Lin, Yin-Yi Chang, Shang-Wen Chen, Jan-Gowth Chang

**Affiliations:** 1Department of Radiation Oncology, China Medical University Hospital, Taichung 404332, Taiwan; meg@livemail.tw (Y.-C.L.); d4615@mail.cmuh.org.tw (J.-A.L.); 2Center for Precision Medicine, China Medical University Hospital, Taichung 404, Taiwan; t92989@mail.cmuh.org.tw; 3Department of Pathology, China Medical University Hospital, Taichung 404, Taiwan; d13463@mail.cmuh.org.tw; 4Department of Laboratory Medicine, China Medical University Hospital, Taichung 404, Taiwan; T32245@mail.cmuh.org.tw (Y.-X.H.); T34752@mail.cmuh.org.tw (S.-J.T.); 5Graduate Institute of Clinical Medical Science, School of Medicine, College of Medicine, China Medical University, Taichung 404, Taiwan; d6375@www.cmuh.org.tw (Y.-C.H.); lsyeh@www.cmuh.org.tw (L.-S.Y.); wei66@iris.seed.net.tw (W.-C.C.); 6Department of Obstetrics and Gynecology, China Medical University Hospital, Taichung 404, Taiwan; sun.vau@msa.hinet.net (W.-C.L.); changyinyi@hotmail.com (Y.-Y.C.); 7School of Chinese Medicine, College of Medicine, China Medical University, Taichung 404, Taiwan; 8Department of Radiology, School of Medicine, College of Medicine, Taipei Medical University, Taipei 110, Taiwan

**Keywords:** cervical adenocarcinoma, chemoradiotherapy, genomic alteration, next-generation sequencing, myeloid cell leukemia-1

## Abstract

This study investigated the prognostic effects of genomic biomarkers for predicting chemoradiotherapy (CRT)-based treatment outcomes in patients with adenocarcinoma (AC) of the uterine cervix. In all, 21 patients receiving definitive CRT were included. In accordance with the International Federation of Gynecology and Obstetrics (FIGO) staging system, 5, 8, and 8 patients were classified as having stage IB3, II, and III disease, respectively. Pretreatment biomarkers were analyzed using tissue microarrays from biopsy specimens. Genomic alterations were examined by next-generation sequencing (NGS). The outcome endpoints were disease-free survival (DFS), distant metastasis-free survival (DMFS), and local relapse-free survival (LRFS). A Cox regression model was used to examine the prognostic effects of the biomarkers and clinical parameters. The presence of myeloid cell leukemia-1 (*MCL1*) gene amplification and a lower immunohistochemical (IHC) marker of tumor necrotic factor alpha (TNF-α) H-score were two prognostic factors for inferior DFS. The four-year DFS was 28% and 68% for patients with or without MCL1 copy number gain, respectively (*p* = 0.028). In addition, MCL1 amplification predicted poor DMFS. A lower tumor mutation number (TMN) calculated from nonsynonymous mutations was associated with lower LRFS. For patients with adenocarcinoma of the uterine cervix receiving definitive CRT, prognostic information can be supplemented by *MCL1* amplification, the TMN, and the TNF-α H score.

## 1. Introduction

Uterine cervical cancer is the second most common cancer in women and the leading cause of cancer death in women in most developing countries [1]. Adenocarcinoma (AC) of the uterine cervix constitutes approximately 10% to 20% of all cervical carcinomas with a trend toward a growing incidence [2,3]. Several risk factors, including obesity, and nulliparity may associate with this increase [4,5]. Screening programs can fail to detect AC because of its deep localization in the endocervical canal. Some studies have suggested that AC and squamous cell carcinoma (SCC) behave differently epidemiologically [2,3,4,5,6,7], and genomic differences exist [8,9]. Additionally, AC patients have inferior outcomes after the same treatment [4,6,10,11,12,13]. At present, most treatment knowledge of cervical AC derives from studies where AC was present in a minority of patients. Hence, there is a need to advance molecular profiling of treatment strategies specifically for AC.

Although chemoradiotherapy (CRT) has been the standard of care for patients with locally advanced cervical cancer, patients with cervical AC primarily treated with CRT have inferior outcomes compared with those with squamous cell carcinoma [4,12,13,14]. As advances in genomic profiling have allowed for the identification of biomarkers of many biological characteristics in tumor cells, biomarkers have a potential role in the design of personalized therapeutic strategies targeting individual tumors. Because there is a great need to identify biomarkers and since there are few investigations of prognostic factors for cervical AC, this study was conducted to investigate the impact of pretreatment genomic markers from next-generation sequencing (NGS) on CRT-based treatment in these patients.

## 2. Results

### 2.1. Treatment Outcomes

In all, 21 patients with newly diagnosed AC of the uterine cervix were included in this study. After a median follow-up duration of 48 months (range, 7–120), 10 patients had no evidence of disease progression. Of the 11 patients with tumor progression, 6 had local recurrence, 3 had distant metastasis, and 2 had both. Furthermore, 8 patients had local residual or recurrent tumors at primary sites, whereas 5 patients experienced distant metastasis.

### 2.2. Genomic Alterations

The genomic alterations were studied by whole exome sequencing (WES) in 16 patients. In the other 5 patients, targeted DNA sequencing was used because of the limited amount of tumor tissue. Nonsynonymous mutations of the individual tumors are summarized in Appendix A. By frequency of occurrence, common somatic mutations included 6 in *PIK3CA*, 5 in *TP53*, 4 in *AR*, 4 in *PTEN*, 4 in *KMT2C*, 3 in *KRAS*, 3 in *ARIDIA*, and 3 in *KMT2D*. The comparative results reported in The Cancer Genome Atlas (TCGA) are shown in Table 1. The absolute number of the nonsynonymous mutations, defined as the tumor mutation number (TMN), ranged from 2 to 71 with a median value of 6. Furthermore, calculation of copy number variations showed *MCL1* amplification in 7 tumors (33%), and *MYC* amplification in 2 tumors (10%).

To understand the associated biological functions of the mutations, the Kyoto Encyclopedia of Genes and Genomes signal pathway was utilized to assume the involved molecular pathway of the mutations. The results are compared with the report from The Cancer Genome Atlas (TCGA) as shown in Table 1.

### 2.3. Predictive Ability of Genomic Alteration, the TMN, and Immunohistochemical (IHC) Biomarkers

All aforementioned immunohistochemical (IHC) biomarkers were retrieved. The predictable IHC biomarkers, clinical parameters, and the TMN and the area under the ROC curve (AUC) are listed in Appendix A. The TNF-α H-score most accurately predicted the occurrence of tumor progression (AUC: 0.155, *p* = 0.007), whereas the TMN had a marginal impact (AUC: 0.259, *p* = 0.062), as depicted in Appendix A. The AUCs for the *TNF-α* H-score with an optimal (12.5) or median (15.0) cut-off were 0.141 (*p* = 0.005) and 0.232 (*p* = 0.038), respectively. Additionally, *MCL1* amplification was associated with cancer recurrence (χ^2^ test, *p* = 0.031).

The sole biomarker that predicted the occurrence of local relapse was the TMN using either continuous values (AUC: 0.197, *p* = 0.023) or the median cut-off of 6 (AUC: 0.216, *p* = 0.033). Furthermore, *MCL1* amplification could predict distant metastasis (χ^2^ test, *p* = 0.011). The association between *MCL1* amplification and treatment outcome is detailed in Appendix A.

None of the other IHC biomarkers, including the hypoxia, cell adhesion, or immunogenicity biomarkers, appeared to be prognostic for tumor progression in this cohort. Therefore, *MCL1* amplification, the TMN, and TNF-α, combined with clinical parameters including age, stage, maximum tumor dimension, brachytherapy schemes, and pretreatment hemoglobulin, were selected for the multivariate Cox regression model for survival analyses.

### 2.4. Prognostic Factors for Disease-Free Survival (DFS), Local Relapse-Free survival (LRFS), and Distant Metastasis-Free Survival (DMFS)

As summarized in Table 2, Cox regression analysis indicated that *MCL1* amplification (*p* = 0.012, hazard ratio (HR) = 10.07, 95% confidence interval (CI) = 1.65–61.49) and the *TNF-α* H score (*p* = 0.024, HR = 0.88, CI = 0.79–0.98) were prognostic factors for disease-free survival (DFS). As depicted in Figure 1, the 4-year DFS of patients with or without *MCL1* amplification was 28% and 68%, respectively (*p* = 0.028), and among patients who had tumors with a TNF-α ≥15 or <15, it was 85% and 22%, respectively (*p* = 0.003).

Cox regression analysis disclosed that the a TMN ≥6 was the sole predictor of a poor local relapse-free survival (LRFS) (*p* = 0.014, HR = 19.27, CI = 1.65–225.50), whereas the MCL1 gain had a marginal impact on LRFS. The 4-year LRFS in patients with tumors with TMN ≥6 or <6 was 90% and 41%, respectively (*p* = 0.006, Figure 2). In addition, *MCL1* amplification was the major determinant for a low distant metastasis-free survival (DMFS) (*p* = 0.021, HR = 22.94, CI = 1.61–148.57). The 4-year DFS of patients with or without amplification was 41% and 92%, respectively (*p* = 0.021, Figure 3).

In univariate or multivariate analysis, age, tumor dimension, pretreatment hemoglobulin, and brachytherapy schemes were not identified as prognostic factors for the aforementioned endpoints.

### 2.5. Quantitative Differences in Clinical Parameters and Biomarkers between Tumors with or without MCL1 Amplification

Using the Mann–Whitney U test, an association analysis was carried out to investigate the quantitative differences in the IHC and clinical parameters between tumors with or without *MCL* amplification. As shown in Figure 4, *MCL1* amplification was found in patients presenting with lower pretreatment hemoglobulin levels (11.6 ± 1.6 vs. 7.9 ± 3.2, *p* = 0.02), or tumors with a lower programmed death-ligand 1 (PD-L1) combined positive score (CPS) (20.1 ± 18.0 vs. 4.4 ± 7.3, *p* = 0.031). Although the mean MCL1 H-scores were higher in tumors with *MCL1* copy number gain, no statistical significance existed. The mean MCL1 H-scores in tumors with or without *MCL1* amplification were 156.4 ± 50.4 vs. 127.1 ± 39.7, respectively (*p* = 0.19). None of the other clinical parameters or IHC biomarkers were associated with *MCL1* amplification.

### 2.6. Validation Results in TCGA

The *MCL1* copy number variation was calculated based on data from the 29 aforementioned samples in TCGA. As depicted in Appendix A, certain patients with *MCL1* amplification were likely to have inferior overall survival compared with those without amplification, but there was no statistically significant difference between the two groups (*p* = 0.39).

## 3. Discussion

In cervical cancer, several biomarkers for radiotherapy-based treatment have been validated by patient survival and recurrence data [15,16].These biomarkers fall into categories according to biological function including hypoxia, cell proliferation, cell adhesion, immunogenicity, and evasion of apoptosis [15]. Given that patients with cervical AC have worse outcomes than those with SCC [4,12,13,14], it is imperative to explore the biological mechanisms specifically for AC patients. Except for the report from TCGA [9], no comprehensive genomic studies for this specific tumor type are available for clinical practice. This work was a pilot study to combine genomic alterations with wide-ranging IHC biomarkers in predicting the outcomes of patients with locally advanced disease receiving definitive CRT. Herein, we disclosed that high *MCL1* copy number gain was correlated with DFS and DMFS. Although the validation analysis from TCGA samples failed to discriminate the overall survival in the same patient setting, this discrepancy may possibly be attributed to the different treatment modalities, or follow-up durations between the two cohorts. Additionally, an objective comparison may be blurred by the sample size. Before considering a novel therapeutic strategy for cervical AC, therefore, further validation studies are required to confirm the findings.

We previously identified the *MCL1* H-score, an anti-apoptotic member of the *BCL2* family of apoptosis-regulating proteins, as an independent factor for cancer death in patients with cervical AC receiving CRT [17]. The current study verified that *MCL1* copy number gain was also associated with inferior treatment outcomes. *MCL1* overexpression has been reported in some hematological cancers and solid tumors [18]. *MCL1* blocks the progression of apoptosis by binding the pro-apoptotic proteins *BCL2* homologous antagonist killer (*Bak*) and *BCL2*-associated protein X (*Bax*), which are capable of forming pores in the mitochondrial membrane, allowing the release of cytochrome c into the cytoplasm [19,20]. Probably due to the limited sample size in this study, the *MCL1* copy number gain was not directly correlated with the *MCL1* H-score. Because *MCL1* has been shown to be both an intrinsic and acquired resistance factor that limits the efficacy of various antitumor agents for different cancers, this highlights the need to employ effective *MCL1* inhibitors that can be used either as a single agent or in combination regimens [21]. We first disclosed that *MCL1* amplification of a human cancer was associated with lower quantitative values of the PD-L1 CPS and pretreatment hemoglobulin. To date, rare studies have reported the relationship between the *MCL1* copy number and the tumor immunogenic microenvironment. Ishibashi et al. indicated that the knockdown of *PD-L1* in myeloma cells downregulated expression of anti-apoptotic genes (*BCL2* and *MCL1*) [22]. Thus, a direct interplay between *MCL1* gene amplification and modification of *PD-L1* expression merits molecular studies. On the other hand, our data revealed that patients having tumors with *MCL1* amplification had a lower baseline serum hemoglobulin, which is not always correlated well with tumor hypoxia [23]. Given that the other IHC hypoxic markers, a crucial factor of radioresistance, were not significantly related to *MCL1* amplification in our data, it would be interesting to clarify whether the *MCL1* copy number gain was driven by tumor hypoxia, or whether the *MCL1* amplification-related cancer cell survival resulted in tumor hypoxia.

Theoretically, WES allows a correct measurement of the tumor mutation burden (TMB), yet it remains expensive, labor-intensive, and time-consuming. Therefore, several studies have explored the possibility of providing equally accurate and clinically predictive mutation burden quantification from various targeted gene panels [24]. This study did not use a straight calculation of the TMB because WES could not be applied for all tumors. Instead, we calculated the TMN from the targeted cancer panel and found that a lower TMN was associated with inferior local control. To confirm the prognostic role of the absolute somatic mutation number, future studies should compare the correlation between the TMB and TMN for a specific cancer, and compare their impact on CRT outcomes. In addition, it should be clarified whether there are some DNA repair genes such as *BRCA* or *ATM* in tumors with a higher TMN.

TNF-α is a vital cytokine involved in inflammation and immunity, and could function as an endogenous tumor promoter [25]. Conversely, TNF-α in the tumor microenvironment can differentially modulate radiosensitivity through a variety of signaling mechanisms. Bioinformatics analysis of lung cancer patients revealed that higher expression of TNF-α was associated with a low risk of cancer progression [26]. This study first showed that the TNF-α H-score was associated with cancer relapse in patients with cervical AC. Because none of the other clinical parameters or IHC biomarkers were found to be related to the TNF-α H-score, this finding needs to be validated in future studies.

Our study has several limitations. First, this was a retrospective study with a limited sample size in a single institution. External validation studies using an independent dataset are necessary to confirm these findings. Particularly, future studies should enroll patients prospectively and employ a standardized NGS protocol. Second, the precise molecular pathway that *MCL1* amplification or TMN confers to poor CRT-based outcomes could not be clarified through association molecular studies, animal experiments, or clinical trials. In addition, no available data on the specific human papilloma virus types could reduce some prognostic or biological information. Finally, the association between DNA sequencing and the protein product should be investigated to understand the comprehensive molecular mechanism of radioresistance or distant metastasis for these patients. Nevertheless, the strengths of this study include the NGS sequencing and wide-ranging analyses of IHC biomarkers. Our findings provide a hint that future studies can clarify the mechanisms related to the failure of CRT. In addition, this study initiated a pilot step to enable the tailoring of CRT to specific biological characteristics of patients with cervical AC. Our findings disclosed that certain genomic information on the tumors may supplement well-known clinical prognostic factors in predicting CRT-based treatment outcomes. Oncologists could then assess the feasibility of personalized therapy for high-risk patients, such as salvage surgery, dose escalation schemes, and a novel combination therapy.

## 4. Materials and Methods

### 4.1. Study Population

Between July 2009 and December 2015, 42 patients newly diagnosed with cervical AC were screened for this study, and 21 patients with tumor specimens fitting the quality control of the NGS were included in the analysis. All patients had undergone F-18 fluorodeoxyglucose positron emission tomography (PET)/computed tomography (CT) for staging and had received allocated external-beam radiotherapy and intracavitary brachytherapy. The eligibility criteria included patients with stage IB3 to IIIB disease in accordance with the International Federation of Gynecology and Obstetrics (FIGO) [27]. Accordingly, 5, 8, and 8 patients were classified as having stage IB3, II, and III disease, respectively. The median age of our patients was 54 years. The diagnosis of lymph node metastasis was based on PET/CT. None had paraaortic lymph node metastasis. We excluded patients who were diagnosed as having a histological type of adenosquamous carcinoma. This study was approved by the local institutional review board (CMUH107-REC3-008, 9/May/2019). Patient characteristics are listed in Table 3.

### 4.2. DNA Extraction

Formalin-fixed paraffin-embedded (FFPE) tumor tissues were sliced into 10 μm slides and 5 slides were collected for DNA extraction. The slides were treated with deparaffinization solution (QIAGEN, Hilden, Germany) and DNA were extracted using a QIAamp^®^ DNA Mini kit (QIAGEN, Hilden, Germany) according to the manufacturer’s instructions.

### 4.3. Whole Exome Sequencing

A total of 300 ng DNA was used for library preparation using a TruSeq^®^ Exome Kit (Illumina, San Diego, CA, USA) according to the manufacturer’s instructions. Briefly, DNA was mechanically fragmented on a Covaris M220 Focused-ultrasonicator (Woburn, MA, USA). End repair, A-tailing, adaptor ligation, enrichment of DNA fragments, and probe hybridization were then performed. Quality control was performed using an Agilent Bioanalyzer 4200 (Agilent Technologies, Santa Clara, CA, USA) to ensure the library size of 200–400 bp. Samples were subjected to 2 × 150 bp paired-end sequencing using the Illumina NovaSeq 6000 platform (Illumina, San Diego, CA, USA).

### 4.4. Targeted DNA Sequencing

A total of 200 ng DNA was used for library preparation using a QIAseq Targeted DNA Panel (QIAGEN, Hilden, Germany) according to the manufacturer’s instructions. The Human Comprehensive Cancer Panel (Cat. No. DHS-3501Z) was chosen to sequence 275 cancer-related genes. Briefly, DNA was enzymatically fragmented and end-repaired according to FFPE DNA reaction conditions. Then, adaptor ligation, target enrichment polymerase chain reaction (PCR), and universal PCR amplification were performed. Cycling conditions for target enrichment PCR were as follows: 95 °C for 13 min; 98 °C for 2 min; six cycles of 98 °C for 15 s and 65 °C for 15 min; 72 °C for 5 min; and 4 °C for at least 5 min. Cycling conditions for target enrichment PCR were set up according to FFPE DNA conditions. The libraries were subjected to 2 × 150 bp paired-end sequencing using the Illumina NovaSeq 6000 platform.

### 4.5. Data Analysis

Base calling and quality scoring were performed with an updated implementation of Real-Time Analysis on the NovaSeq 6000 system. We used bcl2fastq Conversion Software (v2.20.0.422, Illumina, San Diego, CA, USA) to demultiplex data and convert BCL files to FASTQ files. Sequence reads were processed by read trimming, read aligning, barcode clustering, and gene-specific primer masking. Then, single-nucleotide polymorphism and small insertion or deletion were determined in individual samples using smCounter (v2, QIAGEN, Hilden, Germany) at the default settings (Varscan was used for WES). Variants were annotated using dbSNP, ClinVar, Catalogue of Somatic Mutations in Cancer, and The Cancer Genome Atlas (TCGA) to identify whether they had been seen before. If the variants were of unknown significance (sorting intolerant from tolerant (SIFT)), PolyPhen-2, or functional analysis through hidden Markov models (FATHMM) were used to predict the impact of amino acid changes on protein function. Potential candidates were those predicted to be pathogenic either by FATHMM or by both SIFT and PolyPhen-2.

### 4.6. Definition of Myeloid Cell Leukemia-1 Amplification

We previously showed that the IHC marker of MCL1 H-score was correlated to cancer death in this patient setting. To clarify the impact of copy number variation of the *MCL1* gene on the outcome, we used Samtools (v1.3.2, Wellcome Trust Sanger Institute, Cambridge, UK) to obtain the sequencing depth of each point on chromosome 1 from the bam file, and then calculated the average depth of chromosome 1 as standard copy. The sequencing depth of *MCL1* was compared with six genes near *MCL1* to determine the copy number variation of the *MCL1* gene. An example of tumor *MCL1* amplification is shown in Appendix A.

### 4.7. Immunohistochemistry

As mentioned previously [17], IHC biomarkers, namely endogenous hypoxic (Glut1, CAIX, and HIF-1α), angiogenesis or metastasis (VEGF, c-Met), cell proliferation (EGFR, c-Myc, insulin-like growth factor 1 receptor (IGF-1R)), cell-to-cell adhesion (E-cadherin, Vimentin), evasion to apoptosis (B-cell lymphoma 2 (BCL2), Bax, MCL1), immunogenic or inflammatory biomarkers (programmed cell death protein ligand 1 (PD-L1), tumor necrosis factor-α (TNF-α), calretinin, galectin-9, and chemokine ligand 5 (CCL5)) were analyzed using tissue microarrays from incisional biopsy specimens before treatment. Each tumor was represented by one tissue core on a tissue microarray. Furthermore, 4 µm thick paraffin sections were deparaffinized and microwaved according to standard procedures before being processed for IHC staining.

The microscopic stains were scored by two pathologists blinded to the clinical outcome. Except for PD-L1, IHC results from the aforementioned biomarkers were scored by a semiquantitative approach used to assign an H-score to tumor samples [17,28].

The tumor PD-L1 biomarker was evaluated through IHC staining using the DAKO clone 22C3 pharmDx kit (DAKO, Carpinteria, CA, USA). PD-L1 expression was scored according to the combined positive score (CPS), which is the number of PD-L1 stained cells (tumor cells, lymphocytes, macrophages) at any intensity divided by the total number of viable tumor cells, multiplied by 100 [29].

### 4.8. Treatment

Patient treatment has been described previously [17,30]. All patients were treated with intensity-modulated radiotherapy. The total dose applied to the pelvis was 45 Gy, administered in 25 fractions over a five-week period. Following pelvic irradiation, the bilateral parametrium was boosted from 50.4 to 54 Gy.

After adequate tumor regression, high-dose-rate intracavitary brachytherapy was performed once or twice a week using an Ir-192 remote afterloading technique concurrently with pelvic irradiation or parametrial boosting. Before January 2013, the standard prescribed dose for each session of brachytherapy was 6.0 Gy to Point A, with five sessions. To maximize the therapeutic window, 15 patients after January 2013 were treated with three-dimensional image-based brachytherapy according to the recommendations of the Groupe Européen de Curiethérapie and the guidelines specified by the European Society for Radiotherapy and Oncology [31]. The details of the cumulative dose are summarized in Table 3.

Chemotherapy consisted of weekly 40 mg/m^2^ doses of cisplatin, administered intravenously for a total dose of 60 mg.

### 4.9. Follow-Up

After completion of CRT, patients were regularly followed up every two months for the first year, and every three to four months thereafter. Besides a routine pelvic examination, the serum levels of tumor markers, namely carcinoembryonic antigens, were examined during each follow-up. Additionally, a radiographic examination was performed every six months. Patients presenting with symptoms of central-pelvic recurrence underwent a salvage hysterectomy or pelvic exenteration if feasible. Patients with distant metastasis were treated with systemic chemotherapy.

### 4.10. Statistical Analysis

To examine correlations between the aforementioned parameters and tumor recurrence, receiver operating characteristic (ROC) curves were constructed to evaluate the optimal predictive performance among the various parameters. In addition, the χ^2^ test and Mann–Whitney U test were performed to examine the categorical or continuous variables for predicting clinical outcomes. The outcome endpoints were DFS, DMFS, and LRFS, all of which were calculated using the Kaplan–Meier method. The log-rank test and Cox regression analysis were performed to examine the effects of explanatory variables on these endpoints. The stage, age, lymph node status, maximum tumor dimension, hemoglobulin level, and predictable IHC markers were included for multivariate analysis. Patient survival was measured from the date of initiation of radiotherapy to the last follow-up. Two-tailed tests were used, and *p* < 0.05 was considered statistically significant. All calculations were performed using SPSS, Version 13.0 for Windows (SPSS Inc., Chicago, IL, USA).

### 4.11. External Validation in TCGA

To compare the clinical impact of the *MCL1* copy number between our patients and TCGA (Cervical Squamous Cell Carcinoma and Endocervical Adenocarcinoma, CSEC) cohort, the UCSC Xena platform (UC Santa Cruz Genomics Institute, Santa Cruz, CA, USA) was used [32]. First, we selected the “Gene Expression” and “Copy Number” dataset of the *MCL1* gene in the study, i.e., “GDC TCGA Cervical Cancer”. Then, all missing values of the copy number variation were filtered out. The disease type was defined as “Adenomas and Adenocarcinoma”, sample type was “Primary Tumor”, and primary site was “Cervix uteri”. Thirty-one samples were found, but survival data were missing in two. Thus, 29 patients were stratified with a median value of the copy number.

## 5. Conclusions

For patients with AC of the uterine cervix receiving definitive CRT, prognostic information can be supplemented by *MCL1* amplification, the TMN, and an IHC biomarker of the *TNF-α* H-score. The presence of high *MCL1* gain and lower TNF-α expression were two prognostic factors for inferior DFS. A lower TMN was associated with lower LRFS. External validation studies are required to verify our findings.

## Figures and Tables

**Figure 1 ijms-21-04117-f001:**
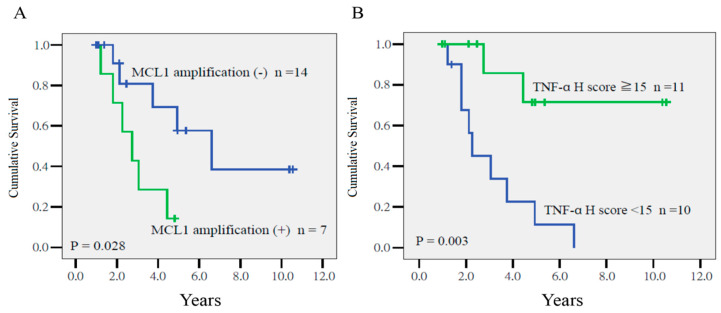
Disease-free survival in patients who had tumors with or without myeloid cell leukemia-1 (*MCL1*) amplification (**A**), and a high (≥15) or low TNF-α H-score (<15) (**B**) (*p* = 0.028 and *p* = 0.03, respectively).

**Figure 2 ijms-21-04117-f002:**
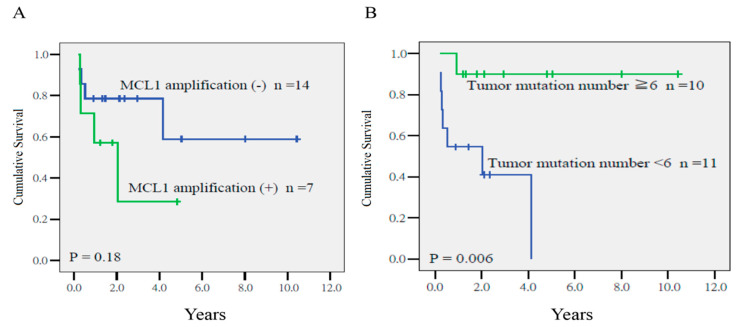
Local relapse-free survival in patients who had tumors with or without *MCL1* amplification (**A**), and a high (≥6) or low tumor mutation number (<6) (**B**) (*p* = 0.18 and *p* = 0.006, respectively).

**Figure 3 ijms-21-04117-f003:**
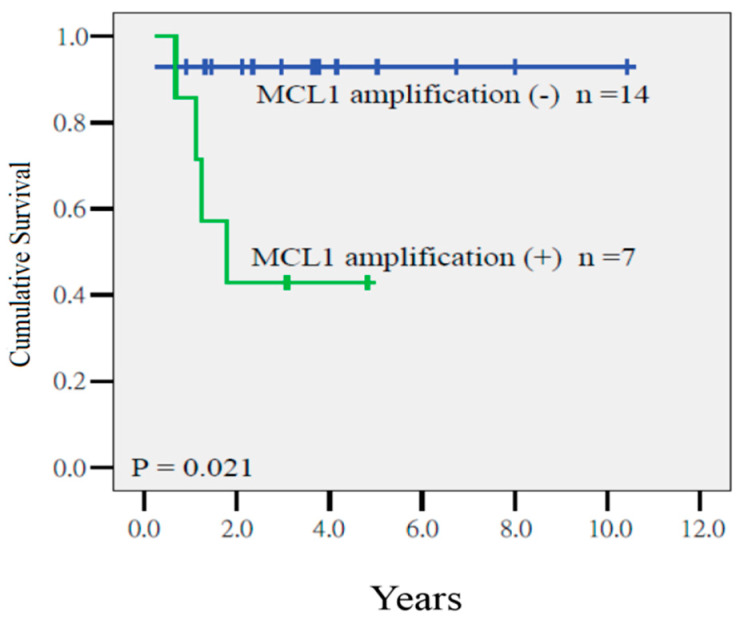
Distant metastasis-free survival in patients who had tumors with or without *MCL1* amplification (*p* = 0.021).

**Figure 4 ijms-21-04117-f004:**
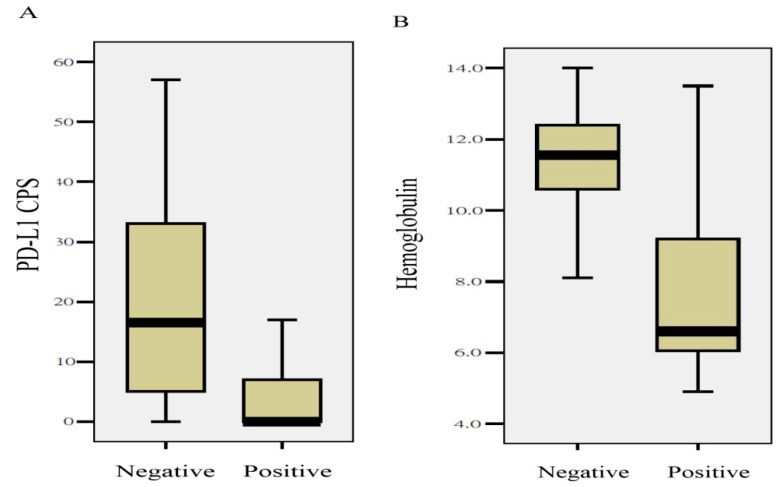
*MCL1* amplification associated with quantitative values of the PD-L1 combined positive score (CPS) (**A**), and pretreatment hemoglobulin (**B**) (*p* = 0.031 and *p* = 0.02, respectively).

**Table 1 ijms-21-04117-t001:** The associated Kyoto Encyclopedia of Genes and Genomes signal pathway and comparative data from The Cancer Genome Atlas (TCGA).

Kyoto Encyclopedia of Genes and Genomes Signal Pathway		Oncogenic Signaling Pathways in TCGA	Frequency of Mutated Gene in TCGA
Category	Number (percentage)		
*RAS*	3/21 (14%)		*KRAS* 7/28 (25%)
*MAPK*	2/21 (10%)		
*ERBB*	4/21 (19%)		*ERBB2* 6/28 (21%)
Combined *RTK/RAS*	5/21 (24%)	63%	
*PI3K-AKT*	6/21 (29%)		*PIK3CA* 9/28 (32%)
*mTOR*	5/21 (24%)		
Combined *P13K*	6/21 (29%)	56%	
*TP53*	6/21 (29%)	19%	
Cell cycle	3/21 (14%)	21%	
TGF-beta	3/21 (14%)	21%	
WNT	2/21 (10%)	14%	
HIPPO	2/21 (10%)	14%	
NOTCH	4/21 (19%)	30%	
APOPTOSIS	3/21 (14%)		
VEGF	3/21 (14%)		

**Table 2 ijms-21-04117-t002:** Multivariate analysis of disease-free survival (DFS), local relapse-free survival (LRFS), and distant metastasis-free survival (DMFS).

Outcome	Variable	*p* Value	HR	95% CI
Disease-free survival	*MCL1* amplification	0.012	10.07	1.65–61.49
	TNF-α H-score (continuous)	0.024	0.88	0.79–0.98
Local relapse-free survival	Tumor mutation number ≥ 6	0.014	19.27	1.65–225.50
	*MCL1* amplification	0.057	4.19	0.96–18.30
Distant metastasis-free survival	*MCL1* amplification	0.021	22.94	1.61–148.57

Abbreviations: HR = hazard ratio; CI = confidence interval.

**Table 3 ijms-21-04117-t003:** Patient characteristics (N = 21).

Variables	Value
Age (year)	median 54 (range, 41–77)
FIGO stage	
IB3	5 (24%)
IIB	8 (38%)
IIIB	2 (9%)
IIIC1	6 (29%)
Maximum tumor dimension (cm)	mean 5.7 ± 1.1 (range, 3.9–8.0)
Pretreatment hemoglobulin (g/dL)	mean 10.3 ± 2.7 (range, 4.9–13.5)
Carcinoembryonic antigen (ng/dL)	mean 36.2 ± 29.3 (range, 0.5–331.7)
External beam radiotherapy (cGy)	
whole pelvis (Gy)	median 45 (range, 39.6–54)
bilateral parametrium boost with central shielding (Gy)	median 54 (range, 50.4–57.6)
pelvic lymph node boost (Gy)	median 64 (range, 60–66)
Brachytherapy	
2-dimensional brachytherapy (6Gy to point A per session for 5 courses) Cumulative EQD2 of point A (Gy_10_)	6
	mean 84.3 ± 7.3
3-dimensional brachytherapy (HR-CTV > 6.5Gy per session for 5 courses) Cumulative EQD2 of D90 of HR-CTV (Gy_10_)	15
	mean 88.1 ± 10.3

Abbreviations: FIGO = International Federation of Gynecology and Obstetrics; EQD2 = equivalent dose in 2 Gy; HR-CTV = high-risk clinical target volume.

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
