# Peer review of "Genomic Biomarkers of Survival in Patients with Adenocarcinoma of the Uterine Cervix Receiving Chemoradiotherapy"

_ijms, 2020, doi:10.3390/ijms21114117_

Round 1

Reviewer 1 Report

The authors in this study evaluate the prognostic effects of genomic biomarkers to predict the outcomes of chemoradioterapy in patients with adenocarcinoma of the uterine cervix. The study is interesting even if it has some limitations such as the small size of the patients, but it could be considered a preliminary study to further investigation. To identify new biomarkers to predict the outcome of the treatment in cancer patients is the most important factor to have personalized therapy. Moreover molecular studies could be very interesting to evaluate the biological mechanism caused by MCL1 amplification.

I suggest to the authors to report the table 1 in the text before the table 2.

Author Response

The order of the tables has been revised accordingly.

Reviewer 2 Report

Authors conducted retrospective study of adenocarcinoma treatment with chemo-radiotherapy. They aimed to find biomarkers for survivals of uterine cervix cancer patients. Authors identified various biomakers associated with prognosis.

Manuscript is clear and well written. I have some minor things.

This manuscript is formatted as abstract, introduction, results, discussion, and materials and methods. It seems this manuscript was originally prepared in the regular format as materials methods before results. Then, I see many abbreviations was not defined in the results sections but in the materials and methods. Please fix this issue and define abbreviations when it shows up in the first time. Examples are below and may not be limited to only them. 

define AC in line 46

define SC C in line 54

define IHC in line 81

TCGA

It is obvious that images need higher resolutions. It was very difficult to see.

Author Response

The images and abbreviation have been revised accordingly.

Reviewer 3 Report

The manuscript by Lin and collaborators describes the findings for several determinations on 21 cases of adenocarcinoma (AC) of the uterine cervix. Cervical AC differ from squamous cell carcinomas (SCC) in frequency, response to treatment and clinical outcome, thus conferring a specific interest in the investigation of genomic and immunohistochemistry characteristics in order to identify possible useful biomarkers. However, the design of the study and the small number of cases included render it difficult to generalize the results.

In particular, the following aspects need to be improved:

1-Study design: this is a retrospective study, and the included cases were newly diagnosed between July 2009 and December 2015; since the number of cases included in another study by the same group during the same period (ref. 17) was 42, a selection has been made for the present study. Please, specify the reason and the criteria used for the selection , and evaluate whether this selection could have introduced any bias. To this regard, this could eventually explain the differences observed with the TCGA validation group (lines 139-140).

2-Treatment: please specify why the treatment regimen was modified in January 2013 (lines 297-298), and how this could impact in the results observed.

3-HPV: both SCC and AC are causally related to persistent infection by high-risk HPV. While in the introduction this is stated (although the statement in lines 45-46 that HPV18 represents one of the reasons related to the increasing frequency of AC is not correct), no data on the specific HPV types involved in the studied cases are then provided. This could give a more complete picture of the carcinomas.

4-Results: a more analytical description of the results on MCL1, with a correlation between amplification and immunohistochemistry H score data, would be useful to better understand the prognostic significance of MCL1 H score (lines 130-131, 163-164).

5-Lines 65-66: the sentence "In summary,....." is not clear and could be deleted.

6-Lines 167-168: the first sentence is not clear and should be rephrased.

7-Line 203: for the statement "..uniform treatment strategies..", please refer to comment 2

8-Lines 207-210: the conclusions are too strong and not completely supported by the results.

Author Response

All of your comments have been revised accordingly or explained one by one. Please see the attached file.

Round 2

Reviewer 3 Report

The authors have responded to my questions and observations, including adequate informations in the manuscript.